

# Characterizations of novel pesticide-degrading bacterial strains from industrial wastes found in the industrial cities of Pakistan and their biodegradation potential

Noreen Asim[1,*], Mahreen Hassan[2,3,*], Farheen Shafique[4,5], Maham Ali[6], Hina Nayab[7], Nuzhat Shafi[5], Sundus Khawaja[8] and Sadaf Manzoor[9]

[1] Division of Genomics and Bioinformatics Institute of Biotechnology and Genetic Engineering, The University of Agriculture Peshawar, Peshawar, Khyber Pakhtunkhwa, Pakistan
[2] Department of Molecular Biology and Biotechnology, University of Sheffield, Sheffield, Yorkshire, United Kingdom
[3] Microbiology, Shaheed Benazir Bhutto Women University, Peshawar, KPK, Pakistan
[4] Department of Biomedical Science, University of Sheffield, Sheffield, Yorkshire, United Kingdom
[5] Department of Zoology, University of Azad Jammu and Kashmir Muzaffarabad, Muzaffarabad, Azad Kashmir, Pakistan
[6] Department of Zoology, University of Peshawar, Peshawar, Khyber Pakhtunkhwa, Pakistan
[7] Institute of Biological Sciences, Sarhad University of Science and Information Technology, Peshawar, Khyber pakhtunkhwa, Pakistan
[8] Department of Biotechnology, University of Azad Jammu and Kashmir Muzaffarabad, Muzaffarabad, Azad kashmir, Pakistan
[9] Department of Statistics, Islamia College University, Peshawar, Khyber Pakhtunkha, Pakistan
* These authors contributed equally to this work.

Corresponding author
Mahreen Hassan,
muhassan1@sheffield.ac.uk,
microkust@gmail.com

## ABSTRACT

**Background**. Lack of infrastructure for disposal of effluents in industries leads to severe pollution of natural resources in developing countries. These pollutants accompanied by solid waste are equally hazardous to biological growth. Natural attenuation of these pollutants was evidenced that involved degradation by native microbial communities. The current study encompasses the isolation of pesticide-degrading bacteria from the vicinity of pesticide manufacturing industries.

**Methods**. The isolation and identification of biodegrading microbes was done. An enrichment culture technique was used to isolate the selected pesticide-degrading bacteria from industrial waste.

**Results**. Around 20 different strains were isolated, among which six isolates showed significant pesticide biodegrading activity. After 16S rRNA analysis, two isolated bacteria were identified as *Acinetobacter baumannii* (5B) and *Acidothiobacillus ferroxidans*, and the remaining four were identified as different strains of *Pseudomonas aeruginosa* (1A, 2B, 3C, 4D). Phylogenetic analysis confirmed their evolution from a common ancestor. All strains showed distinctive degradation ability up to 36 hours. The *Pseudomonas aeruginosa* strains 1A and 4D showed highest degradation percentage of about 80% for DDT, and *P. aeruginosa* strain 3C showed highest degradation percentage, *i.e.*, 78% for aldrin whilst in the case of malathion, *A. baumannii* and *A. ferroxidans* have shown considerable degradation percentages of 53% and 54%, respectively. Overall, the

degradation trend showed that all the selected strains can utilize the given pesticides as sole carbon energy sources even at a concentration of 50 mg/mL.

**Conclusion**. This study provided strong evidence for utilizing these strains to remove persistent residual pesticide; thus, it gives potential for soil treatment and restoration.

# INTRODUCTION

Pesticides are hydrocarbons widely used in agriculture and households to control variety of pests (*Bhatt et al., 2021*). Pesticides were introduced in Pakistan during the 1960s in the period of green revolution, and during the 1980s its applied quantity rose from 665 tons to 46,000 tons in the year 2000 (*Syed et al., 2014*). Pesticides are categorized into four types: organochlorines, organophosphorus, carbamates and pyrethroids. Among these, organochlorine pesticides were the primary class with high toxicity, hydrophobicity, and persistent nature. Many organochlorine pesticides have been banned. However, pesticides like chlorinated hydrocarbons such as DDT, aldrin, and dieldrin are still ignorantly used, and a significant number of obsolete pesticides have been stockpiled in developing countries (*Arrebola et al., 2015*). Among all the synthetic pesticides, organophosphate is the more stable and less toxic class. Therefore, their wide structural variety and diverse chemical properties enable them for multiple uses. Derivatives of these organo-chemicals are also controversial because they also have persistent effects on the surroundings when applied for a more extended period (*Jayaraj, Megha & Sreedev, 2016*) and even within the organism after exposure. Organochlorine and organophosphate, even at low concentration, are toxic and carcinogenic to mammals. They can affect the nervous system, immune system, respiratory system, reproductive function, and hormonal balance, and cause serious illnesses in invertebrates and vertebrates. Accumulation of these pollutants in soil or water bodies has been the reason for the loss of many marine and sea animals. Dramatic consumption would lead to the integration of these pesticides in the food chain, which ultimately accumulate in the tissue of consumers, causing deleterious effects (*Kim, Kabir & Jahan, 2017*). Additionally, wide ranges of water and terrestrial ecosystems have been polluted with these compounds. The eradication and safe removal of such pollutants from the environment is essential, and industries related to pesticides are the main culprits (*Al Hattab & Ghaly, 2012*). Advanced countries have treatment facilities for safe disposal of effluents, but these issues are rarely tackled in underdeveloped and developing countries, particularly in small scale industries due to their unaffordability (*Mishra et al., 2021*). Biodegradation and detoxification with the help of native microbes are promising approaches for toxic waste removal and environmental remediation. Microbial communities on exposure to new compounds generate new enzymes with the emergence of genetic mutations to survive. Hence, native microbes adapt to new surroundings.

Furthermore, it is easy to extract genes from microbes possessing degradation ability, which can be expressed in other hosts by genetic engineering (*Ye, Dong, & Lei, 2018*; *Regueiro et al., 2015*; *Bhatt et al., 2020a*; *Bhatt et al., 2020b*; *Bhatt et al., 2020c*). Many lower income countries do not have any facility to regulate, manage, and dispose of the obsolete persistent organic pollutants. Thus, implementation of cost-effective waste management procedures like bioremediation, phytoremediation and bioaugmentation is needed to restore the natural resources (*UNEP, 2018*).

Biodegradation of pollutants through natural attenuation is the most preferable, cost effective and safest solution. But the availability, effectiveness, and popularization of certain microbial communities for treatment of specific pesticides are a significant issue and are under consideration by examining and isolating the native strains from such contaminated (effluent) areas that combat these toxicities (*Cycoń, Mrozik & Piotrowska-Seget, 2017*). Biodegradation through microbes is the most effective way to make our environment pesticide free. For their sustenance, microbes utilize toxic substances such as carbon, nitrogen, and phosphorus, etc. (*Bhatt et al., 2021*). Currently, microbial degradation is considered as most thorough and extensive method for biodegradation. For this purpose, bacteria, fungi, and algae are all used in different ways, but the bacteria are very common. In the past, many scientists have worked in search of microbes that could degrade pollutants to teir non-toxic states (*Huang et al., 2020*). Bacteria and fungi have been utilized as whole-cell biocatalysts in a variety of biotechnological processes, not only for manufacturing various products but for the development of biotransformation/ biodegradation reactions also (*Birolli, Lima & Porto, 2019*). Many bacterial and fungal genera such as *Aerobacter, Acinetobacter, Agrobacterium, Aspergillus, Bacillus, Clostridium, Desulfuromonas, Enterobacter, Flavobacterium, Fusarium, Klebsiella, Lactobacillus, Pasteurella, Pseudomonas, Pleurotus, Phlebia sp., Penicillium, Rhizopus, Sedimentibacter, Streptococcus*, and *Staphylococcus* were identified in degradation of popular pesticides. Although many biodegrading microbes have been extensively studied for their pesticide degrading ability, many scientists are still searching for better microbes with high and specific efficacy (*Kumar et al. 201, 2018*; *Bhatt et al., 2021*). There are many bacteria which can degrade a variety of xenobiotic compounds quite efficiently under controlled laboratory conditions, but they failed to do so in a natural environment (*Arora et al., 2017*).

Although the use of most organo-pesticides is also banned in Pakistan, their residues still exist and have accumulated in biotic and abiotic resources (*Mazzoni et al., 2020*).

The current study aims to isolate and identify novel bacterial strains from the vicinity of pesticide manufacturing industrial sites that could use pesticides like DDT, aldrin and malathion as their nutritional source.

## Contribution

Our study's uniqueness was to collect samples specifically from the unexplored points of pesticide wastes to isolate the possible native bacterial strains. We also observed the extent of bacterial chemical utilization capacity, especially for DDT, aldrin and malathion. The study gives an insight into the soil that was chemically polluted with these pesticides and could be restored for further cultivation purposes. After confirming their presence, further

chemical tests and morphological studies were conducted to improve our understanding of these bacterial strains that can further be utilized to treat chemically polluted land and water.

## MATERIALS & METHODS

### Sampling and management

Samples were obtained from effluents of five industrial sites, mainly located at Faisalabad and Lahore in Pakistan. Water and soil samples were randomly collected from industrial drainage sources and garbage/disposal points, respectively, where the possibility of the presence of effluents was higher. The soil and water samples were collected with the consent of the landowner. No animal or plant species were harmed during the collection process. Standard procedures were used to obtain sludge samples (*Ghanem, Orfi & Shamma, 2007*), from wastewater near industrial area carrying effluents from the pesticide manufacturing plants (*Farhan et al., 2012*). All the reagents and chemicals were analytical grade purchased from Merck, Germany, and Sigma-Aldrich USA. The pesticides (malathion, DDT and aldrin) with >99% purity were obtained from Sigma-Aldrich, USA. Synthetic oligonucleotides were provided by the Integrated DNA Technologies (IDT), USA for this study. Physical samples of microbial isolates used in this study were permanently deposited at National Institute of Biotechnology and Genetic Engineering (NIBGE) Faisalabad, Pakistan

### Sample processing and enrichment

To maintain the biological activity of the soil microflora, the samples were obtained from contaminated areas in sterile polythene bags kept at 4 °C. On the day of collection, the soil samples were further treated by mixing 10 g of soil sample in a 100 mL of autoclaved water. The mixture was then kept at room temperature (30 °C) in an orbital shaker for 24 h at 100 rpm. The enrichment culture method was acquired to isolate pesticide biodegrading bacterial strains. To improve the likelihood of detecting pesticide-degrading native microflora, samples were first supplemented with commercial-grade pesticides (*Khan et al., 2016*). From each water and soil sample, 10 mL were inoculated into 100 mL of fresh MSM broth in 250 mL Erlenmeyer flasks containing 20 mg/mL selected pesticides as the sole source of carbon and phosphorus. The flasks were incubated for seven days at 150 rpm under ambient conditions. The cultures were gradually acclimated to increase the three applied pesticides ranging from 20–50 mg/mL at weekly intervals. At about a final concentration of 50 mg/mL of different pesticides, the pesticide tolerant cultures were subjected to further degradation study. Media without inoculation was run as a control. The experiments were performed in triplicate for authentication (*Jiang et al., 2019*).

### Isolation of biodegrading strains

For the isolation of pesticide degrading bacteria, the nutrient agar media plates were first inoculated with one mL of each enriched culture and then incubated at 37 °C till colonies appeared. Lone colonies were picked and streaked aseptically on solid media plates followed by incubation at 37 °C for 48 h. Separate colonies were sub-cultured onto nutrient agar

plates till pure cultures were obtained. The strains from pure culture were then inoculated into 100 mL MSM broth which contained 50 mg/mL of selected pesticides added as the sole source of carbon or phosphorus, or both carbon and phosphorus sources to determine their degradation ability at different time intervals (2, 6, 8, 12, 16, 20, 24, 30, and 36 h) at 30 °C. Negative controls were always run to ensure that mentioned sources in the media were not contaminated. The measurement of the absorbance was done by using an Agilent Cary 5000 Ultraviolet-to-Visible to-Near-Infrared (UV-Vis-NIR) spectrophotometer (Agilent Technologies, Petaling Jaya, Malaysia) (*Roy et al., 2018*). The cultures with the highest degradation ability were stored in glycerol as nutrient-agar slants at 4 °C.

$$\text{Degradation Rate \%} = \frac{\text{Initial OD} - \text{Final OD}}{\text{Initial OD}} \times 100$$

### Identification and characterization of isolates

The identification and characterization of biodegrading isolates were carried out using morphological, biochemical, and molecular tests. The preserved isolates showing maximum degradation activity were picked and grown on the nutrient medium through spread plate method followed by culturing at 37 °C for 48 h as described by *Jiang et al. (2019)*. A phase-contrast microscope Axiovert (Zeiss model MC-80) was used to examine the colony features as well as the morphological structure of the cells. Then, strains capable of degradation were further tested through gram staining for the identification. The biochemical tests included catalase test, gelatinase test, starch hydrolysis test (amylase test), casein hydrolysis test, oxidase test, Indole production test, Methyl red test, Citrate utilization test, Nitrate reduction test and Urease test.

### Molecular characterization

The isolated strains' DNA was extracted using the Cetyltrimethylammonium Bromide (CTAB) technique (*Orek, 2018*). The RT PCR (QIAGEN) was used to conduct the 16S rDNA amplification process and phylogenetic analysis. The DNA obtained from the isolated strains was utilized as a template for PCR amplification using 16S rDNA primers. The forward primer for the bacterium was FD1 (AGAGTTTGATCCTGGCTCAG 20 bp), and the reverse primer was rP1 (ACGG(ACT)TACCTTGTTACGACTT, 23 bp). The PCR products were sequenced by Macrogen Inc., Seoul, Korea. The Basic Local Alignment Search Tool (BLAST) was run to analyze sequences (*Altschul et al., 1990*). The 16S rRNA gene sequences of selected strains were acquired from GenBank and matched with the gene sequences of our isolates using CLUSTALX. A distance matrix was constructed using the aligned sequences (*Jukes & Cantor, 1969*). Following the generation of 500 bootstrap sets, MEGA X version 10.2.6 was used to construct a phylogenetic tree using the neighbour-joining technique (*Stecher, Tamura & Kumar, 2020*).

### Growth rate study

To check the efficacy of growth of pesticide degrading microbes in MSM broth, the growth curve was observed.

In a sterile conical flask holding 100 mL of MSM broth, an aliquot (1 mL) of bacterial suspension was added (*Ghanem, Fahad & Abdulghafoor, 2012*). The MSM composition (w/v) was 0.1% $NH_4NO_3$, 0.1% NaCl, 0.15% $K_2HPO_4$, 0.05% $KH_2PO_4$, 0.01% $MgSO_4.7H2O$, 0.0025% $FeSO4$, 1% glucose and incubated in a shaking incubator at 150 rpm using WiseCube Fuzzy System (model WIS-20) (*Al-Thukair & Karim, 2016*) at various temperatures (15 °C, 30 °C and 45 °C). The wavelength was set at 600 nm (OD600) to check cell density at different time intervals (2, 6, 8, 12, 16, 20, 24, 30 and 36 h) using an Agilent Cary 5000 Ultraviolet-to-Visible to-Near-Infrared (UV-Vis-NIR) spectrophotometer (Agilent Technologies, Petaling Jaya, Malaysia). The optimum pH of the growth media was adjusted to pH 7. The growth curve was constructed by determining the optimum culture time of each biodegrading strain at different temperature. All the experiments were done in triplicate, and a negative control with no microbial inoculation was carried under the identical circumstances.

## Statistical analysis

The normality of the data was tested using the Pearson's correlation tests for all of the data obtained. Statistical analysis was done using MS Excel 16.0.

## Approval of field permit

Field experiments were approved by Research committee of the University of Agriculture Peshawar, Pakistan (Project number:27/FCPS/AUP) and National institute of biotechnology and genetic engineering (NIBGE), Faisalabad-Pakistan.

# RESULTS

## Isolation of pesticide-degrading bacteria

Through enrichment, many indigenous pesticide-tolerant bacteria with pesticides as their only carbon source were identified from contaminated soil and industrial effluents. From these enrichment cultures, 20 morphologically diverse strains were isolated on nutrient agar plates containing DDT and aldrin and malathion pesticides, respectively. These strains' pesticide degradation ability was checked on MSM agar plates containing pesticides (50 mg/mL). The six bacterial isolates used in the degradation experiments. Isolates 1A, 2B, and 3C came from soil contaminated with DDT (from Industrial city Faisalabad). While isolates 4D, 5E and 6F were from the industrial area of Lahore, Pakistan, with DDT, aldrin and malathion contamination. Of the two locations investigated, samples from the Lahore industrial area had the highest contamination. Pesticide contamination was found in all the soils and effluents utilized to isolate microorganisms. All six isolates were correctly identified using 16S rRNA sequencing. The strain *Pseudomonas* was found to be the most common among the isolates.

## Pesticide degradation activity

Out of 20 bacterial strains from the industrial wastes, only six (1A, 2B, 3C, 4D, 5E and 6F) showed significant degradation against all three pesticides at 50 mg/mL, which was tested. The plates were marked into small segments. The zone of degradation was plotted against the periodic time intervals shown in Figs. 1, 2 and 3 for the pesticides.

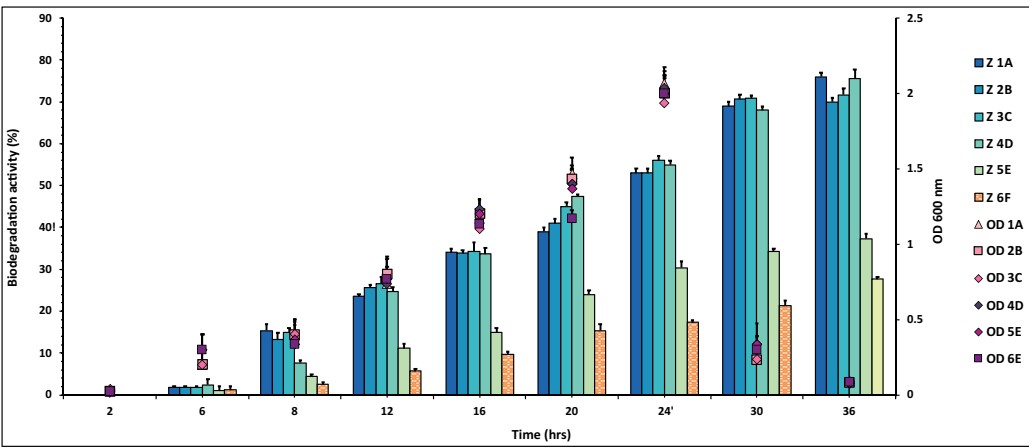

**Figure 1 Graphical representation of percent DDT degradation showed by each bacterial isolate plotted against time.** The experiment was done in triplicate, and errors bars indicates the standard deviation.

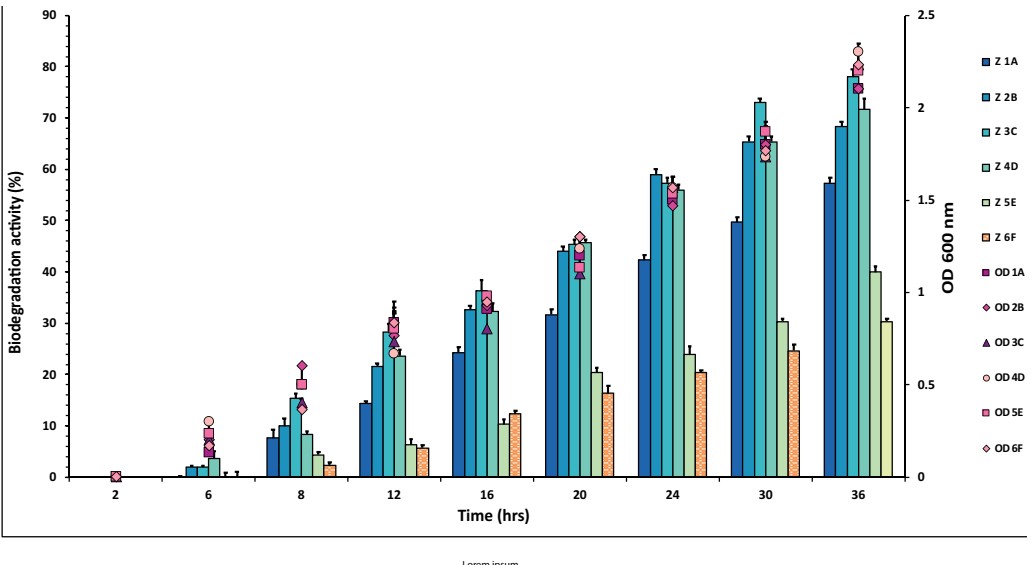

**Figure 2 Graphical representation of percent aldrin degradation shown by each bacterial isolate plotted against time.** The experiment was done in triplicate, and errors bars indicates the standard deviation.

The percentage degradation of DDT pesticide was plotted against time. The isolates 1A, 2B, 3C and 4D showed highest degradation activity of 70–80% during the 36 h while 5E and 6F degraded 28–36% of the same DDT concentration (Fig. 1 and Table S1).

In the case of aldrin, a little variation in percentage values of degradation was observed when plotted against time (Fig. 2 and Table S2). The isolate 3C showed the highest degradation (78%) against aldrin, followed by 2B and 4D with 72% and 68%, respectively. In contrast, isolate 1A showed 58% degradation. The degradation values for the isolates 5E

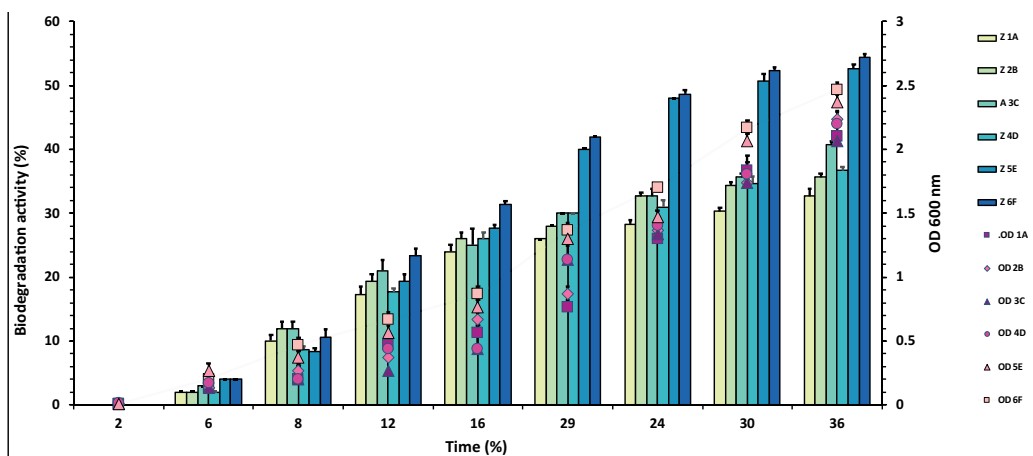

**Figure 3 Graphical representation of percent malathion degradation shown by each bacterial isolate plotted against time.** The experiment was done in triplicate, and errors bars indicate the standard deviation.

and 6F were 30% and 40%, respectively. However, 5E and 6F values were observed to be lower than 1A, 2B, 3C and 4D.

In the case of malathion, a reverse trend was observed, and highest values of 54% and 53% were recorded from F6 and E5 isolates over 1A, 2B, 3C, 4D, where degradation rate was recorded between 30–40%, as shown in Fig. 3 and Table S3.

Overall degradation of the selected bacterial isolates showed that selected locally collected bacteria could easily degrade organophosphates and organo-chlorines. However, the degradation rate can depend on both the type of bacteria and the pesticide composition, which needs to be explored further. These strains could play a useful role in achieving the safe recovery of polluted sites, particularly in a situation where pesticides, chemicals and other undesired toxic substances containing organochlorine and organo-phosphates and their variety of derivatives might persist. These isolates might also be helpful to produce beneficial microbes sensitive to such toxicities. However, the isolated strains had varying effectiveness for specific pesticides. This trend showed that strains might be applied according to the nature of pollutant present in a particular environment for eradicating toxic chemicals.

## Identification and characterization of isolates

The bacterial strains were selected by analyzing their morphology and colony characteristics, indicating the microbial diversity present in the respective collected (environmental) samples. Based on the isolates' physical appearance, the distinguishing yellow, cream, and white pigmented colonies (Fig. 4), a colonial diversity (raised and flat) was observed. When viewed under a phase-contrast microscope, the isolates were appeared with different cellular morphologies. All isolates were mostly rod-shaped, while the isolate 4D exhibited coccus shape as well. Similarly, all isolates were gram-negative and motile except 4D, which is non-motile. None of the isolates contained endospores, and none of the isolates indicated exospores. The isolates 4D, 5E and 6F were circular, while the rest are irregular in shape.

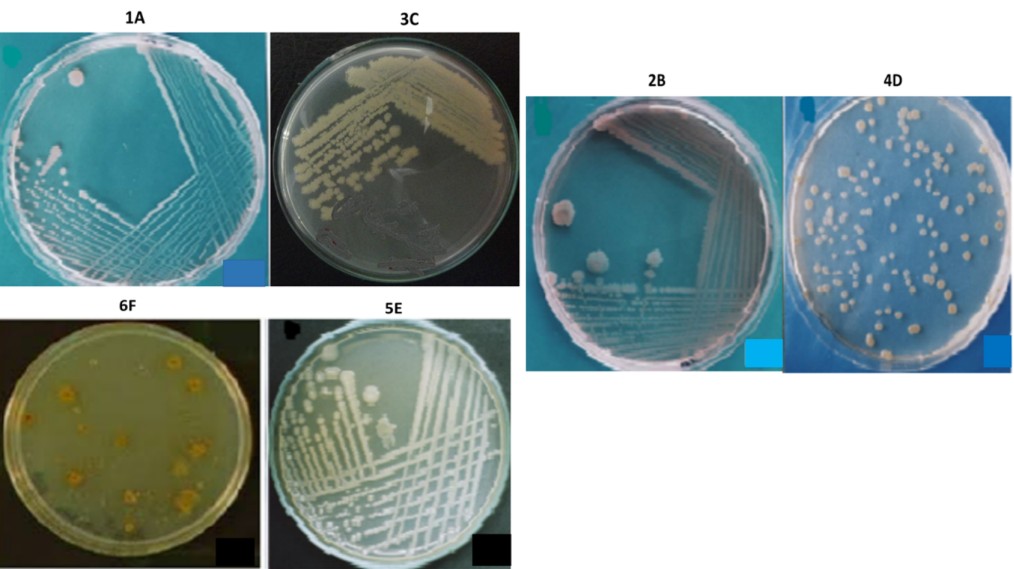

**Figure 4** The appearance of colonies of the bacterial isolates 1A, 2B, 3C, 4D, 5E and 6F on nutrient agar.

**Table 1 Cellular characteristics of biodegrading pesticide isolates.**

| Isolate code | Media used | Shape of cells | Cell size (µm) | Motility | Gram staining | Endo-spores |
|---|---|---|---|---|---|---|
| 1A | Nutrient Agar | Rods | 1.5 | Motile | Negative | Negative |
| 2B | -do- | Rods | 2.0 | Motile | -do- | -do- |
| 3C | -do- | Rods | 2.5 | Motile | -do- | -do- |
| 4D | -do- | Coccus/rod | 1.5 | Non motile | -do- | -do- |
| 5E | -do- | Rods | 1.5-3.0 | Motile | -do- | -do- |
| 6F | -do- | Rods | 2.2 | Mitile | -do- | -do- |

The isolates 2B, 3C, and 4D have undulated colony margins whereas 1A, 5E and 6F had entire margins (Table 1). The selected isolates' cell size was found to be 1.5 µm for 1A and 4D, while the rest of the isolates' cell size ranges from 2.0 to 3 µm, as presented in Table 1.

## PCR method for 16S rRNA amplification and Sequencing

Before sequencing all the DNA samples were amplified under optimized conditions. To obtain a PCR product of 1, 500 base pairs, universal eubacterial rDNA primers fD1 and rP1 were used (Fig. S1). Purified PCR product was sequenced commercially. A partial sequence of the 16S rRNA showed that four isolates had 97–99% homology with *Pseudomonas spp*. Simultaneously, the other two isolates were 97 and 96% identical to *Actinobacter baumannii* and *Acidothiobacillus ferroxidans*, respectively (Table 2).

All these sequences are deposited in the National Center for Biotechnology Information (NCBI) (https://www.ncbi.nlm.nih.gov/).

**Table 2  Sequence identity (%) of the partial 16S rRNA gene sequence of isolates obtained from environmental samples with related microbes.**

| Isolate with codes | Base pairs | Top hit with | % Identity | Gap % | GenBank accession no. |
|---|---|---|---|---|---|
| 1A | 797 | *Pseudomonas aeruginosa strain N7* (MH393215.1) | 99.13 | 0 | HM215142.1 |
| 2B | 796 | *Pseudomonas aeruginosa strain 1816* (MK078036.1) | 97.67 | 0 | HM215143.1 |
| 3C | 808 | *Pseudomonas sp. strain 1AP-CY* (MT084588.1) | 99.02 | 0 | HM215144.1 |
| 4D | 798 | *Pseudomonas aeruginosa strain JU-Ch1* (KF146882.1) | 98.40 | 0 | HM215145.1 |
| 5E | 780 | *Acinetobacter baumannii strain SHRRWY.80* (MT551041.1) | 97.99 | 0 | HM215146.1 |
| 6F | 307 | *Acidothiobacillus ferroxidans* (AY956336.1) | 95.94 | 3 | HM215147.1 |

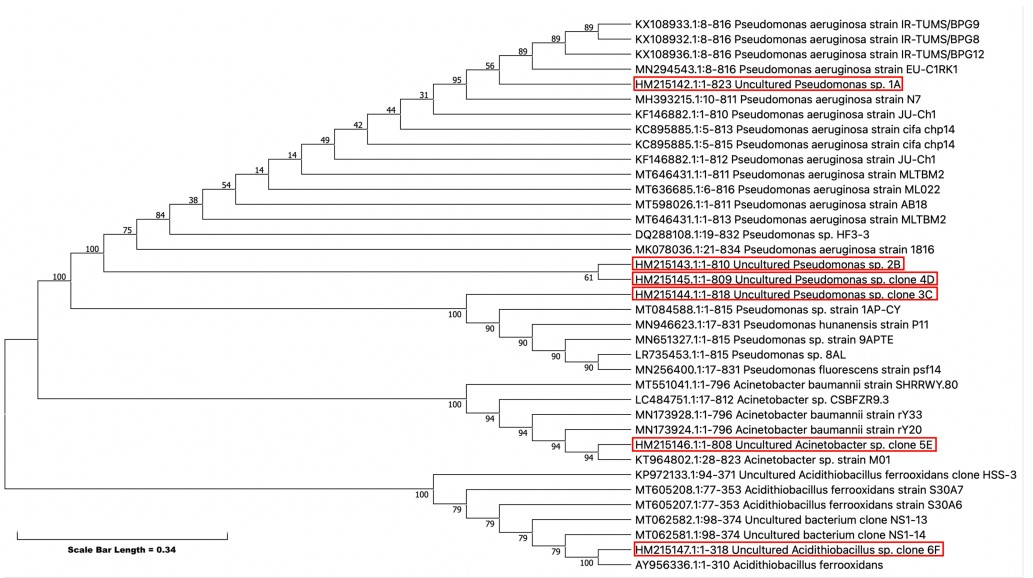

**Figure 5  Evolutionary relationships of taxa -The evolutionary history was inferred using the Neighbor-Joining method (*Saitou & Nei, 1987*).**

## Phylogenetic analysis

Partial 16S rRNA were obtained from the NCBI (http://www.ncbi.nlm.nih.gov) web site, and they were aligned with our isolates to generate a phylogenetic tree (Fig. 5). Phylogenetic analysis showed that all five isolates have evolved from a common ancestor. They have a close relationship among themselves except 5E which shares a common ancestor but is far off from the other isolates in the tree. This showed the diversity among the isolates persists that can tolerate pesticides even at high concentration.

## Estimation of bacterial growth rate

At 600nm, optical density of each isolate was recorded at various intervals *i.e.,* 2, 8, 12, 16, 20, 24, 30 and 36 h, on MSM medium. All the isolates studied in the experiment attained an OD of around 0.7 in broth at 30 °C after 8 h, while it took more than 10 h to reach the OD of 0.7 at other temperatures, so the optimal temperature for these strains was 30 °C, as shown in Fig. 6 and Table S1. 6F showed maximum growth at different temperatures

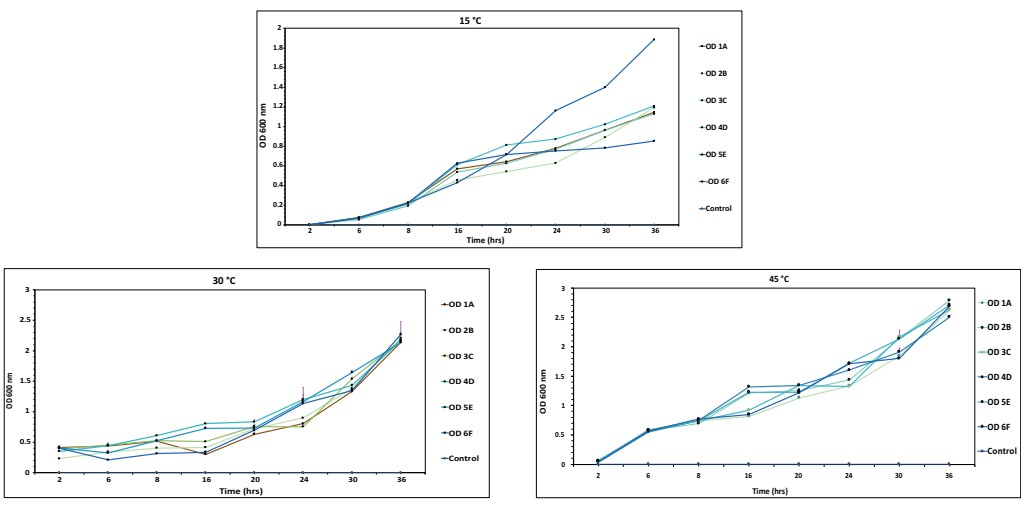

**Figure 6** **Growth curve of isolated strains 1A, 2B, 3C, 4D, 5E and 6F at 15 °C, 30 °C and 45 °C.** The black bars represent standard error.

among all the isolates, while A1 showed the least. Similar results were reported by *Kaya, Aslim & Kariptaş (2014)* and *Murad et al. (2007)*.

## DISCUSSION

The study addresses the isolation and identification of bacteria residing in pesticide-polluted sites. We also observed the extent of bacterial chemical utilization capacity for DDT, aldrin, and malathion. The identification of isolated bacteria was governed by analyzing both phenotypic and genotypic traits. From the current study, microbial biodiversity present in industrial waste was assessed using 16S rRNA analysis which is by far the best genetic marker that could exquisitely identify and illustrate the evolutionary relationship of mixed cultures of environmental and clinical bacterial isolates (*Jovel et al., 2016*; *Lagier et al., 2015*).

In this study, six novel strains were found to have distinct degradation abilities for the three pesticides DDT, aldrin, and malathion. All the isolates used these pesticides as the sole carbon and energy source (*Rani et al., 2018*). The optical density was relatively increasing at the same rate as within the nutrient medium (Fig. 1).

The *P. aeruginosa* strains 1A and 4D showed the highest degradation percentage of about 80% for DDT. Strains 3C and 4D of *P. aeruginosa* also successfully degraded up to 70% of 50 mg/mL DDT as shown in Fig. 1. The *P. aeruginosa* strain has good potential for biodegradation of DDT, also observed in the study done by Sariwati et al. (2018), where *P. aeruginosa* degraded 90% of 5 mM DDT (1.7 mg/mL), a smaller amount as compared to the current study. In another degradation study, (*Wu et al., 2018*) identified *P. aeruginosa* NBRC 3080 and *Bacillus subtilis* NBRC 3009 degraded about 65.97% and 86.44% respectively of 5 mM DDT in 7 days, which was quite similar to our findings. Moreover, other microbes like *Serratia marcescens* NCIM 2919 and *Fomitopsis pinicola* also

degraded 42% DDT in 10 and 7 days of incubation while *P. aeruginosa* addressed in this study utilized pesticide and degraded it in 36 h of incubation (*Grewal et al., 2016*; *Sariwati, Purnomo & Kamei, 2017*; *Sariwati & Purnomo, 2018*; *Rizqi, Purnomo & Kamei, 2021*). This indicates that the strains reported here have a rapid rate of DDT degradation, which could be increased further by the synergic effect of two similar or different species.

For aldrin, all the *P. aeruginosa* strains showed significant activity. The degradation percentage given by *P. aeruginosa* strains 1A, 2B, 3C, 4D, *i.e.*, 58%, 72%, 78% and 68%, respectively, is shown in Fig. 2. Lately, no evidence of aldrin degradation was observed by *P. aeruginosa*. However, other microbes like *P. fluorescens*, White-Rot Fungus (*Pleurotus ostreatus),* and *Paenibacillus sp. IITISM08* have shown the best degradation ability with percentages of 94.8% (conc. 10 mg/L), 100% (conc. 5 mM) and 79% (50 μg/mL), respectively but utilized less concentration as compared to recent isolates that explain the decent activity of our strains (*Purnomo et al., 2017*; *Rani et al., 2018*).

Aldrin and DDT both are organochlorine pesticides, and it was noticed that all the *P. aeruginosa* strains had got reductive de-chlorination ability; therefore, they efficiently degraded these difficult compounds (*Chen et al., 2013*). It is also stated that *P. aeruginosa* possesses hydroxylase enzymes, like oxygenases and reductases, which play their part during the process of degradation (*Rizqi, Purnomo & Kamei, 2021*). DDT is normally metabolized to DDD, DDE, alkane and cycloalkene. But the recent study could not provide the metabolites of degradation. Pseudomonas species are also known for the production of biosurfactant, which apparently caused its increased rate of degradation as compared to other isolates (Sariwati et al., 2018). That might imply their role in malathion degradation to the extent of 30–40%. These observations indicate that *P. aeruginosa* maintains genetic diversity and metabolic flexibility, a phenomenon also addressed by *Chanika et al. (2011)*.

Malathion is an organophosphate pesticide and was degraded efficiently by isolates *A. baumannii* (53%) and *A. ferroxidans* (54%), as shown in Fig. 3. A similar study was done by *Kadhim, Rabee & Abdalraheem (2015)* where *Pseudomonas putida* degraded 47.18% and*Staphylococcus vitulinus* 44.24% in ten days. In contrast, our findings were somewhat similar but were prime in terms of degradation rate. In 2016, Khan et al. identified *Bacillus licheniformis,* which utilized malathion as the sole carbon source and degraded up to 78% within 5 days efficiently (*Kadhim, Rabee & Abdalraheem, 2015*; *Khan et al., 2016*). Whereas in our study, the best degradation percentage was about 54% which is less than the preceding study, but if the time of degradation is exceeded it might show a better result. Malathion, being an organophosphate pesticide, contains phosphorus-ester bonds (P-S bond), which usually proves to be hydrolyzed by esterases and organophosphorus hydrolases. It is observed that esterase is of great importance when microbes use malathion as a sole carbon source. Synthetic pyrethroids like permethrin and malathion are normally used against garden pests. In a previous study, *A. baumannii* strain ZH-14 promotes 100% (50 mg/mL) degradation of permethrin in 72 h by using a similar method (*Zhan et al., 2018*). Co-resistance against both malathion and permethrin is shown by many pests which control unique esterases (*Francis et al., 2017*).

The current study showed that *A. baumannii* and *Acidothiobacillus ferroxidans* utilized malathion as a sole carbon source, which indicates that both the strains may contain

enzyme esterases which could break it down to acceptable metabolites. Furthermore, it is noteworthy that *A. ferroxidans* is mainly involved in bioleaching of elements like sulphur, iron, copper, and heavy metals from waste material. The effective biodegradation of malathion by the strain also implies its capability to utilize phosphate-esters (*Bhatt et al., 2021*) *A. baumannii* and *A. ferroxidans* also degraded DDT and aldrin to the extent of 28–40%. Thus, we infer dynamic catalytic activity in all the isolates with variable degradation rate depending upon pesticide composition, which needs to be explored further.

The current study provides six microbes with varying degradation ability. Almost all the preceding studies highlighted showed degradation at 30 °C in about 7 to 10 days of incubation, whereas the current study identified microorganisms with a higher degradation ability at the same temperature in 36 h. This could be because these organisms were isolated from pesticide polluted lands and waters, therefore, were more thoroughly adapted. This trend showed that particular strains might be applied according to the nature of pollutant present in a specified environment for eradicating toxic chemicals.

### Limitations of the study

Our findings' scope can be further improved by considering the other factors such as organic matter, pH, temperature etc., that might help to pace the specific microbes' availability for biodegradation and their interactive association with chemicals.

Moreover, the study fails to address the extent of degradation of DDT, aldrin, and malathion: whether to their intermediate compounds or a more complete degradation. Furthermore, the enzymes contributing to the degradation ability should be isolated and studied. These deficiencies provide a topic for further investigators in this regard.

## CONCLUSION

Current study found that among all the six strains isolated from polluted pesticide environment, two strains of *P. aeruginosa* (1A and 4D) have shown higher potential ability to degrade 80% of DDT, and *P. aeruginosa* 3C showed the highest degradation percentage, *i.e.,* 78% for aldrin whilst in the case of malathion *A. baumannii* and *A. ferroxidans* have shown considerable degradation percentages of 53% and 54% respectively.

## ACKNOWLEDGEMENTS

This work was supported by the University of Azad Jammu and Kashmir Muzaffarabad, Pakistan and Shaheed Benazir Bhutto Women University, Peshawar Pakistan. The research was supported by University of Agriculture Peshawar; KPK-Pakistan and samples were provided by the National Institute of Biotechnology and Genetic Engineering (NIBGE) Faisalabad-Pakistan, University of Sheffield UK, and Islamia College University Peshawar Pakistan. The manuscript was proof-read by Prof Mike P Williamson from the University of Sheffield UK.

### Funding

This work was supported by the University of Azad Jammu and Kashmir Muzaffarabad, Pakistan and Shaheed Benazir Bhutto Women University, Peshawar Pakistan. The research was supported by University of Agriculture Peshawar; KPK-Pakistan and samples were provided by the National Institute of Biotechnology and Genetic Engineering (NIBGE) Faisalabad-Pakistan, University of Sheffield UK, and Islamia College University Peshawar Pakistan. The funders had no role in study design, data collection and analysis, decision to publish, or preparation of the manuscript.

### Grant Disclosures

The following grant information was disclosed by the authors:
University of Azad Jammu and Kashmir Muzaffarabad.
Pakistan and Shaheed Benazir Bhutto Women University, Peshawar Pakistan.
University of Agriculture Peshawar.
KPK-Pakistan.
National Institute of Biotechnology and Genetic Engineering (NIBGE) Faisalabad-Pakistan.
University of Sheffield UK.
Islamia College University Peshawar Pakistan.

### Competing Interests

The authors declare there are no competing interests.

### Author Contributions

- Noreen Asim and Mahreen Hassan conceived and designed the experiments, performed the experiments, analyzed the data, prepared figures and/or tables, authored or reviewed drafts of the paper, and approved the final draft.
- Farheen Shafique conceived and designed the experiments, analyzed the data, prepared figures and/or tables, authored or reviewed drafts of the paper, and approved the final draft.
- Maham Ali analyzed the data, prepared figures and/or tables, authored or reviewed drafts of the paper, contributed to the final writing of the manuscript, and approved the final draft.
- Hina Nayab conceived and designed the experiments, prepared figures and/or tables, authored or reviewed drafts of the paper, contributed to the final writing of the manuscript, and approved the final draft.
- Nuzhat Shafi analyzed the data, prepared figures and/or tables, authored or reviewed drafts of the paper, contributed to the data collections, and approved the final draft.
- Sundus Khawaja analyzed the data, prepared figures and/or tables, authored or reviewed drafts of the paper, contributed to the manuscript preparation and finalisation, and approved the final draft.
- Sadaf Manzoor analyzed the data, authored or reviewed drafts of the paper, and approved the final draft.

## Field Study Permissions

The following information was supplied relating to field study approvals (i.e., approving body and any reference numbers):

Field experiments were approved by the Research Committee of the University of Agriculture Peshawar, Pakistan (project number: 27/FCPS/AUP) and the National Institute of biotechnology and genetic engineering (NIBGE), Faisalabad-Pakistan.

## DNA Deposition

The following information was supplied regarding the deposition of DNA sequences:

The partial 16S rRNA gene sequence of isolates obtained from environmental samples are available at GenBank: HM215142.1 to HM215147.1.

## Data Availability

The raw data is available in the Supplemental File.

The samples (1A, 2B, 3C 4D, 5E, and 6F) used in this study were physically deposited at the microbial culture library of Industrial Biotechnology Division, NIBGE, Lab G-05, Rack N0. 1-3, Labcool Pharmaceutical Refrigerator (MPR-1410, Sanyo, Japan) under the accession numbers

NA-1APa-009

NA-2BPa-009

NA-3CPs-009

NA-4DPa-009

NA-5EAb-009

NA-6FAf-009, respectively.

For further information you may contact:

Dr. Nasrin Akhtar, Principal Scientist

Industrial Biotechnology Division,

National Institute for Biotechnology and Genetic Engineering (NIBGE),

P. O. Box 577, Jhang Road,

Faisalabad, Pakistan

Phone: +92-41-9201316-Ext. 257

Fax: +92-41-9201322

Web: www.nibge.org

## Supplemental Information

Supplemental information for this article can be found online at http://dx.doi.org/10.7717/peerj.12211#supplemental-information.

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
