# Peer review of "Characterizations of novel pesticide-degrading bacterial strains from industrial wastes found in the industrial cities of Pakistan and their biodegradation potential"

_PeerJ, doi:10.7717/peerj.12211_

## Round 0.1 · original submission · Major Revisions

Your manuscript needs major revisions. The comments from the reviewers are straightforward. Please focus your revisions on improving the discussion section, updating some of the references, and providing more details in the experimental section. When you have made these revisions and others and/or responded to reviewer comments, I would strongly encourage you to review the manuscript carefully for grammar.

·

Basic reporting

Overall paper is good; there are few minor changes that should be made.

Experimental design

Kindly give the reference of the Cetyl trimethyl ammonium bromide (CTAB) method, which is mentioned in the methodology.

Mention which version of the MEGA X software was used.

In the discussion section, it would be better to write short names of the microbes.

Validity of the findings

No comments

Reviewer 2 ·

Basic reporting

The paper is well written and grammatically correct. The figures and the rest of the things are according to the journal format.
Could you replace this reference with an updated one;
Saitou N, N. M. The Neighbor-Joining Method: A New Method for Reconstructing Phylogenetic Trees. Mol. Biol. Evol. 1987. https://doi.org/10.1093/oxfordjournals.molbev.a040454.

Experimental design

The methodology is quite convincing, but my questions are about future perspectives. In conclusion, describe the future perspective, at which stage microbes degrade the DDT and other pesticides. Can the damaged land be recovered?

Validity of the findings

The statistical analysis is acceptable for me.

Additional comments

In acknowledgement, National Institute of Biotechnology and Genetic Engineering (NIBGE) Faisalabad-Pakistan, name should be mentioned .

Reviewer 3 ·

Basic reporting

The study seems interesting, and reporting the novel pesticides degrading strains seems interesting. The paper is well written scientifically. No major grammatical error was found.
Yet there are a few suggestions which could be followed:
• Remove the full stop at the end of the title.
• Remove the colon (:) after the heading “contribution”.
• The heading “Abstract” and the description should not be in the same line.
• Kindly write the manufacturer’s name of the thermocycler used in the methodology.
• After the conclusion's heading, remove the (:) and start with the new line.
• After the Keyword heading, remove (:) and starts with the new line.

Experimental design

I am convinced with the experimental design. The authors have thoroughly explained the research question.

Validity of the findings

The novelty if the finding were fully addressed and informative, and it seems it would help the researchers a lot in the future. The statistical result is quite clear.

Additional comments

The writing style is satisfactory.

Reviewer 4 ·

Basic reporting

The manuscript "Characterizations of novel pesticide degrading bacterial strains from industrial wastes found in the industrial cities of Pakistan and their biodegradation potential" by Noreen Asim and colleagues describes the isolation of 6 different bacterial strains that are able to grow in DDT, aldrin or malathion as only carbon source from different environmental samples. This study provided strong evidence for utilizing these strains to remove persistent residual pesticides; thus, it gives potential for soil treatment and restoration.

Experimental design

The presentation and details are not sufficient.

Validity of the findings

The poor discussion of the results.

Additional comments

The manuscript "Characterizations of novel pesticide degrading bacterial strains from industrial wastes found in the industrial cities of Pakistan and their biodegradation potential" by Noreen Asim and colleagues describes the isolation of 6 different bacterial strains that are able to grow in DDT, aldrin or malathion as only carbon source from different environmental samples. This study provided strong evidence for utilizing these strains to remove persistent residual pesticides; thus, it gives potential for soil treatment and restoration. Overall, this work is interesting but the presentation and details are not sufficient. Some comments and suggestions are listed as follows for improvement.

Major comments:
1- The poor discussion of the results. Author just shows the great amount of results that they have achieved, but they did not use them to develop an interesting discussion which could supplement to earlier studies on biodegradation processes carried out by pure cell cultures.
2- Errors in grammar and language editing. Authors are responsible for preparing their papers in correct English language. The manuscript requires substantial grammatical revisions in its present form and it should not be accepted for publication unless both the technical and grammatical revisions have been made successfully and the English has been polished.
3-Many of the references have been superceded. More modern ones are required.

Specific comments:
1.The word "novel" is used in the title of this manuscript. How do you reflect it?
2.Keywords lack of representativeness and need to be re-summarized.
3.In the Introduction section, it is too long and verbose, and needs further refinement. In addition, many of the references have been superceded. The authors should add more recent references.
4.“The current study encompasses the isolation of organochlorine (DDT and Aldrin) and organophosphorus (malathion) degrading bacteria from the vicinity of industries involved in producing biochemical pesticides.” This conclusion is confusing. Please revise it.
5.Acinetobacter baumannii has been reported to degrade other pesticide in previous study (doi: 10.3389/fmicb.2018.00098). Why the authors did not compare degrading efficiency of strain 5E with that of other strains of A. baumannii?
6.Line 36, “Aldrin” should be changed to “aldrin”. “Malathion” should be “malathion”. Please check the format throughout the paper.
7.The unit format should be uniform, such as line 50 “mg/ml”, line 180 “mg/mL” and line 410 “ mg L−1” and so on. Please carefully check it throughout the manuscript.
8.Line 135, “could use pesticides as their nutritional source”. Please explain clearly which pesticides can be used.
9.37℃ is too hot for most environmental bacteria. Why use 37 ℃ to screen bacteria in line 187?
10.Can Figure 4 be replaced with a high-definition electron microscope image?
11.What is the mean about “do” in Table 1
12.What is the optimum pH and rotation rate for these bacteria?
13.Line 172 and 179, ‘100 r/min’, ‘150 rpm’, The expression of the unit needs to be unified.
14.Line 186, Are you sure it's 1 ml culture?
15.Line 189, ‘48 hrs’, For standardization, it is recommended to use ‘hours’ or ‘h’.
16.Line 233, 15, 30 and 40oC.
17.In the Materials and methods section, please provide methods and equipment for the determination of three pesticides (DDT, aldrin, malathion).
18.Figure 1, 2, 3 and 6, need to increase the resolution of these pictures. I suggest that the author make these figures into broken-line figures to better express the results of the experiment.
19.Figure 4, need to increase the resolution of these pictures.
20.Figure 5, phylogenetic trees need to be redrawn to be more standard.
21.Line 412, [32][33]???
22.Line 421, [18]?
23.References: Many of the references have been superceded and more modern ones are required. The following papers belong to this topic, I hope they are useful for the authors. doi: 10.1016/j.chemosphere.2016.12.129; doi: 10.1016/j.jhazmat.2020.124927; doi: 10.1080/07388551.2020.1853032; doi:10.1016/j.chemosphere.2020.128827; doi: 10.1016/j.envres.2020.110660; doi: 10.3389/fbioe.2021.632059; doi: 10.1186/s12934-021-01556-9; doi:10.3389/fmicb.2019.01453; doi:10.1016/j.biortech.2017.12.007.

---

## Round 0.2 · accepted · Accept

Thank you for addressing reviewer comments and revising your manuscript.

·

Basic reporting

This article is according to the standards of journal. I recommend to publish this article.

Experimental design

It is original, relevant and sufficient details are provided.

Validity of the findings

Conclusion are well stated and linked to original research.

Additional comments

I have found this article suitable and relevant for publication.

Reviewer 2 ·

Basic reporting

no comment

Experimental design

no comment

Validity of the findings

no comment

Additional comments

After going thoroughly through paper, I am completely satisfied with basic reporting, experimental design, and validity of findings. Moreover, my previous comments have also been addressed properly. So, I have no further comments to add up and should be accepted.

Reviewer 4 ·

Basic reporting

The authors have considered all comments raised by the reviewers and revised the manuscript accordingly based on these comments. The revision is fine and can be accepted for publication in current form.

Experimental design

The experimental design is good.

Validity of the findings

The findings is interesting.